# The Effect of Germinated Sorghum Extract on the Pasting Properties and Swelling Power of Different Annealed Starches

**DOI:** 10.3390/polym12071602

**Published:** 2020-07-18

**Authors:** Hesham Alqah, M. S. Alamri, A. A. Mohamed, S. Hussain, A. A. Qasem, M. A. Ibraheem, I. A. Ababtain

**Affiliations:** Department of Food Science and Nutrition, King Saud University, Riyadh 11456, Saudi Arabia; heshamfrnd@gmail.com (H.A.); alamri@ksy.edu.sa (M.S.A.); shhussain@ksu.edu.sa (S.H.); akaram@ksu.edu.sa (A.A.Q.); ibraheem@ksu.edu.sa (M.A.I.); babtain@ksu.edu.sa (I.A.A.)

**Keywords:** α-amylase, susceptibility, starch, viscosity, swelling power, extract

## Abstract

Starches were extracted from chickpea (C.P.), corn (C.S.), Turkish bean (T.B.), sweet potato (S.P.S.), and wheat starches (W.S.). These starches exhibited different amylose contents. The extracted starches were annealed in excess water and in germinated sorghum extract (GSE) (1.0 g starch/9 mL water). The α-amylase concentration in the GSE was 5.0 mg/10 mL. Annealing was done at 40, 50, and 60 °C for 30 or 60 min. The pasting properties of annealed starches were studied using Rapid Visco-Analyzer (RVA), in addition to the swelling power. These starches exhibited diverse pasting properties as evidenced by increased peak viscosity with annealing, where native starches exhibited peak viscosity as: 2828, 2438, 1943, 2250, and 4601 cP for the C.P., C.S., T.B., W.S., and S.P.S., respectively, which increased to 3580, 2482, 2504, 2514, and 4787 cP, respectively. High amylose content did not play a major role on the pasting properties of the tested starches because sweet potato starch (S.P.S.) (22.4% amylose) exhibited the highest viscosity, whereas wheat starch (W.S.) (25% amylose) had the least. Therefore, the dual effects of granule structure and packing density, especially in the amorphous region, are determinant factors of the enzymatic digestion rate and product. Swelling power was found to be a valuable predictive tool of amylose content and pasting characteristics of the tested starches. The studied starches varied in their digestibility and displayed structural differences in the course of α-amylase digestion. Based on these findings, W.S. was designated the most susceptible among the starches and S.P.S. was the least. The most starch gel setback was observed for the legume starches, chickpeas, and Turkish beans (C.P. 2553 cP and T.B. 1172 cP). These results were discussed with regard to the underlying principles of swelling tests and pasting behavior of the tested starches. Therefore, GSE is an effortless economic technique that can be used for starch digestion (modification) at industrial scale.

## 1. Introduction

Starch molecular structure contrasts by botanical source. The X-ray patterns of starches such as corn and wheat have an A-type crystal within the granule, while tubers exhibit B-type. Unlike the B-type, the A-type crystal is a closely packed arrangement of double helices which is more stable thermodynamically. The temperature during the growth of the plant could influence starch composition (ratio of amylose to amylopectin) and structure [1]. Endothermic transition is observed by DSC when starch is heated in an appreciable amount of water, which is termed gelatinization. The gelatinization temperature is correlated with amylopectin located in the crystalline region of the granule [2]. Starch annealing is the process by which starch slurry is heated at a temperature higher than glass transition and lower than gelatinization [3]. Physical changes take place in annealed starch such as increase in gelatinization temperature and sharp gelatinization profile. The gelatinization parameters of wheat starch can dramatically increase when annealed at 50 °C for 72 h [3]. With respect to what happened theoretically to the structure of the annealed starch, little explanation was given in the literature, but one explanation proposed winding of A-chain of amylopectin ends and the linear alignment of amylopectin double helices within the crystalline region of the granule [4]. Therefore, annealing is ordinarily supposed to improve the crystalline and amorphous lamellae by optimizing the crystalline order of the intact starch granule, but it does not increase the helical structure of amylopectin. C-type starch is a mix of B- and A-type starches. Transformation of B-type starch to more stable A-type starch can be achieved by moisture heat treatment. 

Granular (native) starch digestion (hydrolysis) is not as fast as gelatinized starch whose crystallinity has been essentially damaged; amylose and amylopectin became more accessible to enzymes and not restricted by the double helices or amylose-lipid complexes. Bacteria, fungi, plants, and human α-amylase are common endo-enzymes known to hydrolyze α-(1-4) bonds in a random fashion, thereby decreasing the molecular weight of starch molecules. Previous studies have shown that the action of human α-amylase on starches of different botanical origin occurred in various digestion kinetics pathways and produced diverse degradation products [5].

The characteristics of native starch granules that regulate the location, rate, and extent of hydrolysis by α-amylase are not yet known. Nonetheless, the size and shape of granules are controlling factors. Access of α–amylase to its substrate and the release of reaction products are influenced by many factors such as granule integrity, crystallinity, and porosity, amylose to amylopectin ratio, structural, phosphate content (in tubers), lipids on the surface of starch granules, and starch degradation products such as dextrin that have been shown to inhibit α–amylase [6]. Therefore, starch susceptibility to digestion by α-amylase or amyloglucosidase depends on granular structure, the method of preparation, and those molecules bound to the granule surface. 

The factors that contribute to the differences in the kinetics of α-amylase interaction with starch granules are still not well defined. In addition, the changes that occur in starch granules structure during heat processing of foods make starch digestion enzymology puzzling. The variations of α-amylase interaction with starch are mainly accredited to inherent distinctions in the granule structure of diverse botanical species in addition to food processing [7,8].

At small scale, we may picture the surface of a starch granule to differ significantly between starch granules according to origin, from fairly smooth, assembled from small blocklets fixed on an amorphous matrix, to a rougher surface rising from large crystalline blocklets extended from an amorphous structure [9,10]. It is clear that how some examples of enzyme kinetic data can reveal information about the surface structural properties of starch [11,12].

The objectives of this work were to determine the effect of crude extract of germinated sorghum seeds on the physicochemical, pasting, and swelling power of wheat, chickpea, sweet potato, corn, and Turkish beans starches. This study includes the effect of annealing on the physical and pasting properties of the isolated starches. Pure α-amylase enzyme was used parallel to the crude extract so that to assist in monitoring the extract effectiveness. The outcome of this work will allow for using economical crude (un-purified) α-amylase extract in the starch industry for starch physical modification or syrup production.

## 2. Materials and Methods

The raw material was purchased from a local supermarket in Riyadh (Saudi Arabia). The sweet potato used was Salem variety grown in Altayef area of the western Saudi Arabia in the winter, whereas wheat was a hard red winter wheat grown in the north part of Saudi Arabia, the Hail region, in the winter. The chickpea and Turkish bean were purchased from the local market (Riyadh, Saudi Arabia). Corn starch was donated by ARASCO Company (Riyadh, Saudi Arabia). Analytical grade sodium hydroxide, acetic acid, and *Aspergillus* fungal α-amylase (EC3.2.1.1) were purchased from Sigma-Aldrich (St Louis, MO, USA). The centrifugation step of starch isolation was done using Beckman Centrifuge (Beckman JXN, Brea, CA, USA).

### 2.1. Starch Isolation

#### 2.1.1. Chickpea and Turkish Bean Starch Isolation

A slurry was prepared by mixing whole meal of chickpeas or Turkish beans in distilled water (50/50; *weight*/*volume*) in a heavy-duty blender for 5 min (B. Braun Melsungen, AG, Hessen, Germany) and passed through 200 mesh sieves and centrifuged at 2000× *g* for 15 min at 10 °C. The top layer on the precipitate was removed and the white substance at the bottom of the bottle was suspended in distilled water and centrifuged using the above-mentioned conditions. This procedure was repeated five times to obtain pure white starch (Scheme 1). The starch was air-dried at room temperature (30 °C) using acetone, ground in a coffee grinder, and stored in sealed glass bottles at 4 °C for further analysis [13].

#### 2.1.2. Wheat Starches Isolation

Starch was isolated using a method reported by Kasemsuwan et al. [14]. The dough ball was made by mixing flour and water, and the dough was washed with distilled water. The washing water was filtered through a 200 mesh sieve and the through was centrifuged three times at 3000*×*
*g* at 10 °C each time the precipitate was re-suspended in water and the top layer was removed. The isolated starch was dried with acetone, grinded in a coffee grinder, and stored in glass bottles at 4 °C for further analysis.

#### 2.1.3. Sweet Potato Starches Isolation

According to the method of Sit et al. [15], sweet potato tubers were thoroughly washed, peeled, and cut into small pieces. Slurry was prepared by blending the cut tubers in distilled water (50:50 *w*/*v*) for 3 min using an auxiliary kitchen mixer (St 553 Benson Rd, San Antonio, TX, USA). The slurry was filtered through a muslin cloth and the overs were re-blended and filtered in the same way. The starch in the filtrate is allowed to settle for one hour at room temperature and the top liquid was discarded. The precipitate was re-suspended in distilled water and centrifuged at 2000*×*
*g* for 15 min at 10 °C. After centrifugation, the top layer was removed and the white matter at the bottom of the bottle was reconstituted in distilled water and centrifuged three times until pure white starch was obtained. The isolated starch was air-dried and stored in airtight jars at 5 °C.

### 2.2. Amylose Content

Amylose content of the isolated starches was determined according to the method of Williams et al. [16]. The method is based on weighing 0.1 g of starch dry basis, 1.0 mL ethanol, and 9.0 mL NaOH (1.0 M). The mixture was placed in a boiling water bath for 10 min and cooled to room temperature. To 5.0 mL mixture, 1.0 mL acetic acid (1.0 N) plus 2 mL iodine solution (2.0 g of potassium iodide + 0.2 g of iodine diluted to 100 mL with distilled water) and the absorbance (A) was read at 620 nm, then the percent amylose content was calculated using: 3.06 × A × 20.

### 2.3. Sorghum Germination

Germination sorghum was prepared according to Afify et al. [17] with minor modification. Sorghum seeds were germinated at 24 °C and 25% moisture for 4 days, dried, stored at room temperature, ground, and stored at 4 °C. To 10.0 g of the germinated sorghum flour, 40.0 mL of distilled water were added and stirred with magnetic stirrer for 15 min. The slurry was filtered through Whitman 40 and centrifuged for 10.0 min at 2000*×*
*g*. Fresh extract was prepared daily.

### 2.4. Annealing of Starch in Germinated Sorghum Extract (GSE)

Starch (30.0 g) was weighed in a glass jar and 270 mL of distilled water was added to obtain a starch:water ratio of 1:9 (*w*/*v*). To the slurry, 0.1 mL and 1.0 mL of extract were added and one set of samples was annealed without GSE. The slurry was placed in a shaking water bath at 40, 50, and 60 °C for 30 or 60 min with constant agitation at 100 rpm (shaking water bath, Julabo, Germany). The digested slurry was centrifuged to remove the digestion product, the supernatant was discarded, and the precipitate was air-dried. The centrifugation process was repeated three times by adding water to the precipitate for each sample. The dried starch was passed through a 250 µm sieve and stored in at −20 °C for further analysis.

### 2.5. Pasting Properties

Rapid Visco Analyzer (RVA) (Newport Scientific, Sydney, Australia) was used as described by Alamri et al. [18], where starch (2.8 g dry basis) was weighed in a RVA canister and distilled water was added to complete the total weight to 28 g. Extract was added to the starch slurry at 0.1 and 1.0 mL and hand stirred and placed in water bath at 40, 50, and 60 °C for 30 or 60 min. The two different concentrations of GSE was used to make sure that significant measurable effect was done on starch pasting properties. After annealing in the water bath, the samples were analyzed by RVA and the parameters measured include peak viscosity (cP), setback (cP), final viscosity (cP) and pasting temperature.

### 2.6. Gel Texture

Gel texture was determined according to Sandhu and Singh [19] using the Brookfield Texture Analyzer CT3 (Brookfield Engineering Laboratories, Inc., Middleboro, MA, USA). All starch gels obtained from the rapid viscosity analyzer were poured into glass beakers (25 mL) and stored overnight at room temperature. Double compression test was performed using plastic cylinder (45 Perspex Cone, 432-081) attached to crosshead moving at speed of 70 mm/min in both up and down directions. The following parameters were measured: hardness, cohesion, and adhesion.

### 2.7. Swelling Power

The swelling power was measured according to the method described by Waliszewski et al. [20] with slight modifications. Starch slurry was prepared by adding 9.0 mL of water to 0.5 g of starch and vortexed for 30 s (WI). The suspension was heated in a water bath at 70 °C for 30 min with continuous stirring. The mixture was cooled to room temperature and centrifuged at 2000*×*
*g* for 15 min. The precipitate (W2) was weight and the swelling power was measured by the following formula: Swelling power = (W2 − W1)/weight of starch (*g*/*g*).

### 2.8. Estimation of the Amount of α-Amylase in Extract

The α-amylase activity in the GSE was estimated using the effect of pure α-amylase on the pasting properties of the tested starches. Therefore, 1.0% (5 mg/10 mL) α-amylase (EC3.2.1.1) was prepared and 0.1, 0.2, 0.3, 0.4, and 0.5 mL were added to 2.8 g of corn starch. Likewise, 0.1 mL of germinated sorghum extract (GSE) was added to 2.8 g of corn starch. Both corn starch samples were placed in a RVA aluminum can and the final weight (28 g) was completed with distilled water. The slurry was annealed for 30 min and the test was done in the RVA as mentioned above.

### 2.9. Statistical Analysis

All experiments were done in triplicate. Experimental data were analyzed using analysis of variance (ANOVA) and were expressed as mean value ± standard deviation. Duncan’s multiple range test was conducted to assess significant differences among experimental mean values (*p* < 0.05). SAS Foundation 9.2 for Windows was used to analyze the data (SAS Institute, Inc., Cary, NC, USA).

## 3. Results and Discussions

The amylose content (%) of wheat (W.S.), chickpea (C.P.), sweet potato (S.P.S.), Turkish beans (T.B.), and corn starch (C.S.) was 25.0% ± 0.07%, 24.0% ± 0.09%, 22.6% ± 0.06%, 20.9% ± 0.06%, and 20.4% ± 0.08%, respectively. Native starches exhibited peak viscosity as: 2828, 2438, 1943, 2250, and 4601 cP for the C.P., C.S., T.B., W.S., and S.P.S., respectively. Annealed starches without the addition of germinated sorghum extract (GSE) exhibited different pasting properties (Table 1). It was obvious how S.P.S. had the highest peak viscosity and T.B. had the lowest peak viscosity. Although all tested starches showed increase in peak viscosity after annealing regardless of time or temperature, the least increase was recorded for C.S. and the highest for tuber starches (C.P. and T.B.). The limited increase in peak viscosity indicates that the amorphous region of the C.S. granule did not undergo significant molecular arrangement, which is the main effect of annealing on the granules, since annealing as a process is described as heating starch in appreciable amount of water above glass transition and below gelatinization temperature. Under these conditions, starch granules undergo structural transformation, where the amorphous regions reorganize and the overall granule crystallinity increases [3,21,22]. Hydrothermal treatments of starch could suppress granule swelling, retard gelatinization, alter starch gel structure, and increase gel hardness. Therefore, under annealing conditions, C.P. and T.B. starches underwent the most structural rearrangement, especially in the amorphous region, which was reflected on the utmost increase in peak viscosity among the tested starches after annealing (Figure 1). However, W.S. granule was slightly influenced by annealing due to the small increase in peak viscosity and complete liquefaction at 60 °C despite the high amylose content. Higher annealing temperature did not change the ranking of the peak viscosity of the tested starches and S.P.S. stayed the most heat resistant. With respect to the lowest viscosity, C.S. had the least at 40 °C and T.B. at 50 °C. Corn starch appeared to be more sensitive to annealing time since it exhibited the least viscosity after 60 min at 40 and 50 °C without GSE.

During germination of sorghum, the activity of α-amylase increases evidently by the changes that occur during starch gelatinization, such as drop in the peak viscosity. After introducing GSE, the highest viscosity was recorded for the S.P.S. regardless of GSE concentration or annealing temperature, while the lowest was W.S. The viscosity of the remaining starches was in between S.P.S. and W.S. This indicates S.P.S. resistance to α-amylolysis under annealing conditions. At 60 °C, W.S. was completely gelatinized without GSE presence. In addition, C.S. and T.B. were the most heat sensitive when annealed at 40 and 50 °C, but after adding GSE, W.S. was the most sensitive and exhibited the least viscosity. What connects C.S. and T.B. is the similar amylose content despite their botanical origin. All starches, regardless of annealing time, exhibited lower viscosity at higher GSE. The enzyme attack on starch granules increases water absorption, which causes faster granule rupture, leading to lower viscosity. The opposite of this mechanism is when granules resist enzyme attack, which in turn prolongs rupture; consequently, granules maintain swelling for a longer time, and the viscosity subsequently increases. Based on this data, W.S. is the most susceptible among the tested starches due to the lowest viscosity, whereas S.P.S. was the least susceptible because of the highest viscosity followed by C.P. These two starches (S.P.S. and C.P.) did not have the highest amylose content among the tested starches. Variations in the degradation degree by α-amylase was observed among starch samples, indicating that subtle differences in the structure of starch granules with similar amylose content can influence α-amylolysis and the mechanism of hydrolysis.

This occurrence is due to the ratio of amylose to amylopectin, as well as the degree of interaction between starch chains within the amorphous domains in the native granules as shown by XDR [23]. In addition, amylose, which is located in the amorphous region of the granule, serves as a supporting and stabilizing structure which makes the outer layer more rigid and crowded and less penetrable with water and limited swelling. Therefore, the absence of amylose or rigid amylose structure facilitates for the entry of α-amylase, leading to amylolysis and lower peak viscosity [24]. Therefore, S.P.S. granule structure represented by tight entanglement of amylose and amylopectin molecules is the reason for the heat and amylolysis resistance. Concerning W.S., the reason for the high susceptibility to amylolysis and heat could be a loose structure in the amorphous region that assisted for easy water and enzyme penetration, causing fast degradation of the granules. Conversely, C.P. starch has 24% amylose versus 25% for W.S., yet it has much higher amylolysis and heat resistance compared to W.S., indicating a tighter amorphous region. Researchers found the following order of decreasing resistance of native starch granules towards α-amylase: high amylose corn starch > potato > rice > wheat > common corn > tapioca > waxy corn [25,26]. The same authors reported similar hydrolysis rate of α-amylase from different sources such as salivary or pancreatic. These findings are not in agreement with the work reported here because in their ranking, the high amylose starch was the most resistant to enzyme attack, unlike W.S. tested here. However, the disagreement can be attributed to the variation in granule structure rather than amylose content or the enzyme source. Since the action of α-amylase is similar regardless of its source, the reason could be attributed to the difference in the packing of the amorphous region.

Therefore, granule structure and compactness of the amorphous region are the main cause for the difference between the data presented here and the literature [25,26]. Certainly, there were some exceptions at 50 °C and 1.0 mL GSE regardless of annealing time; C.S. and C.P. exhibited higher viscosity, 1836 and 1698 cP, respectively, than S.P.S. (1478 cP). Reports indicated that, at 50 °C, salivary α-amylase hydrolyzes granular corn starch 88% faster than the rate at which it hydrolyzes the dissolved starch form (gelatinized) [27]. Therefore, susceptibility is temperature and time dependent because these parameters have effects on the starch granule structure as well as the enzyme activity. The rate of drop in viscosity as a function of GSE level was more noticeable for W.S. (86.5%) followed by S.P.S. (72.3%) at 40 and 50 °C. Conversely, at 60 °C, W.S. was liquefied with or without GSE, whereas C.P. starch exhibited the highest drop in viscosity at 1.0 mL GSE (Table 1). These data are good evidence for the significantly high susceptibility (*p* < 0.05) of W.S. to α-amylase compared to the other tested starches followed by C.P. Generally, starches with higher viscosity (less susceptible) exhibited higher breakdown without GSE, but by adding more GSE, the breakdown has increased due to the action of α-amylase that continued to degrade the starch fractions after the rupture of the granules.

In the most part, annealed C.P. exhibited the highest setback (2553 cP), whereas C.S. (799 cP) and S.P.S (958 cP) exhibited the least setback; however, in the presence of GSE, W.S. had by far the least setback (568 cP) (Table 2). These two starches, W.S. and C.P., contain the highest amylose content, 25% and 24%, respectively, amongst the tested starches. Moreover, C.P. is the second most attacked by α-amylase after W.S. and S.P.S. (Table 2). Low setback indicates low gel strength which is due to reduction of amylose chain length by α-amylase action. It has been reported that the product of the action of α-amylase on starches from diverse botanical origin varied in degradation patterns, kinetics pathway, rates, and extent of hydrolysis, which can be attributed to the structure of the granule of the native starch. Obviously, the size and shape, integrity, crystallinity, porosity of granules, and amylose to amylopectin ratio are likely the most controlling factors that facilitate access of the enzyme to the granule and determine the structure of the digestion product [5,28,29]. The texture of the gel after α-amylase digestion is affected by the extent of enzyme activity and the structure of the native granule. The low setback is due to the weak amylose network formed after starch gelatinization and cooling. During annealing at 40 °C without GSE, W.S. exhibited setback in the middle among the tested starches, but at 60 °C annealing, it has very limited setback, which indicates no gel formation due to liquefaction. Given that W.S. was the most susceptible to α-amylase, the low setback is due to the low amylose chains length (due to amylolysis), which limits the hydrogen bonding because of proximity. Thus, the gel network was weak or did not occur at all. Conversely, the utmost setback of annealed C.P. is an indication of strong amylose retrogradation and firmer gel due to longer amylose chains and strong hydrogen bonding. In spite of change in annealing temperature, GSE level, and time, C.P appear to maintain setback lead compared to other starches (Table 2). The setback ranking of the annealed starches was C.P. > T.B. > W.S. > S.P.S. > C.S. and annealed in GSE was C.P. > T.B. > S.P.S. > C.S. > W.S. Although S.P.S. was the least susceptible, it ranked before the most susceptible (W.S.) with respect to setback. This is another indicator that granule structure or compactness is the basis for protection against α-amylase attack and not the amylose content. Therefore, W.S. was easily attacked by the enzyme despite the high amylose content which indicates looser granule structure, but C.P. granule with similar amylose content appeared to have a more compact granule structure.

The final viscosity followed the same pattern as the setback where annealed W.S. ranked last (2896 cP) and C.P. first (5108 cP). Starch samples experienced drop in final viscosity at different rate at excess GSE, where C.P. and W.S. had the highest final viscosity drop the most in GSE, 95% and 78% for W.S. and C.P., respectively. The drop was sustained at the 60 min annealing as well, which indicates higher sensitivity to the enzyme degradation after gelatinization. The pasting temperature (PT) ranking of the native starch was: W.S. > S.P.S. > T.B. > C.S. > C.P., but after annealing without GSE, the ranking was W.S. > T.B. > C.S. > S.P.S. > C.P. Starch pasting after annealing at lower temperature indicates that annealing created porous granules, whereby quicker water uptake which leads to earlier pasting. This ranking did not match any of the other parameters except for W.S. swelling which supports the idea of the increased porosity after annealing. After annealing, the PT of most of tested starch samples stayed the same except for slight drop of W.S. from 84.1 to 72 °C, and slight increase for C.P. from 71 to 73 °C.

Since the enzyme activity was not determined on the GSE, the following was done to estimate the amount of the enzyme in the GSE. RVA runs with pure α-amylase was compared to GSE. The pasting profiles of the samples with the GSE and the pure enzyme were overplayed on top of each other in order to match the GSE profile with the profile of the known fungal α-amylase enzyme concentration. The results indicated that by adding 0.1 mL of GSE, the profile matches the profile of the sample with 0.5 mL of the fungal α-amylase. Therefore, the concentration of the enzyme in the GSE was 5 mg/10 mL.

The swelling power ranking of the native (un-annealed) starches was C.P. > S.P.S.> C.S. > T.B. > W.S., but after annealing at 40 °C, the ranking was W.S. > C.S. > C.P. > T.B. > S.P.S. It is obvious how annealing significantly (*p* < 0.05) increased the swelling power of W.S. from the least to the most swollen (Table 3). Annealing at 40 °C reduced the swelling of native S.P.S. from 5.49 to 3.01 (*g*/*g*) which can be attributed to increase in amylose packing in the amorphous region by closing the space between molecules and slowing the water uptake. Therefore, swelling power of S.P.S. is temperature dependent. The swelling power of annealed W.S. ranked first (6.86 *g*/*g*) with no GSE addition regardless of annealing time or temperature and S.P.S. ranked last with an average of 2.2 (*g*/*g*) Table 3). This was consistent with the lowest viscosity of W.S. and the highest viscosity of S.P.S. compared to the other starches (Table 3). Slow swelling allows for holding the water (controlled water uptake), thus maintaining viscosity for a longer time until granule rupture is followed by a drop in viscosity (breakdown). However, low viscosity occurs when W.S. granules uptake water fast and rupture sooner. This was also true when the pasting temperature of W.S. decreased at higher GSE. The swelling power ranking remained the same when samples were annealed in GSE, except for at 50 °C at 60 min, where C.P. starch granules swelling range was 5.01–5.66, W.S. range was 4.60–5.24, and S.P.S. range was 1.98–2.58 (*g*/*g*). The different behavior of the starches in GSE at 50 °C was also observed earlier during the discussion of the pasting properties, (Table 1) indicating higher α-amylase activity at this temperature toward C.P. granules rather than W.S. Good correlation between starch granule swelling and peak viscosity is expected because swelling is the starting step of starch peak viscosity, therefore, peak viscosity is a product of granule swelling. Starch swelling power and swelling volume of selected wheat starches were significantly (*p* < 0.01) positively correlated with starch paste peak viscosity [30]. For the most part, C.P. and C.S. exhibited similar swelling while T.B. and S.P.S. were similar. This allows for grouping the tested starches to C.P., W.S., and C.S.; and S.P.S. and T.B.; whereas native starches grouping was C.P., C.S., and S.P.S.; and T.B. and W.S. Along these lines, starches behaved differently with respect to gelatinization properties and swelling power, but longer annealing time appeared to prolong T.B. degree of swelling as well as the peak viscosity (Table 1 and Table 3). With respect to the amylose content, the data presented here are in agreement with reports in the literature where high amylose content does not always correlate negatively with granule swelling, given that W.S. had the most amylose content yet had the least swelling. Therefore, granule structure and the compactness of the amorphous region of the granule is what makes W.S. to uptake water fast. Amylose content and granule size did not appear to be associated with the swelling power of some starches; however, for other starches, high amylose content correlated negatively with the swelling power [31]. Percent crystallinity was found to have direct effect on starch swelling power and gelatinization parameters as well. Lopez-Rubio et al. [32] showed that the increase in the molecular order is favored by the action of hydrolytic enzymes, which explains why α-amylase can digest native starch granules faster than gelatinized starch. Therefore, amylose/amylopectin ratio was a major factor that influences both swelling and pasting characteristics of starches used in this study, as well as how amylose molecule in the amorphous region entangle. The data presented here provided evidence for the different behavior of starches from different botanical source towards enzymatic hydrolysis due to their diverse structural characteristics. In addition, starch annealing permitted for a greater accessibility of enzymes to the amorphous and the crystalline regions of W.S. granule which promotes significant changes in gelatinization mechanism during enzyme digestion. This action supports the idea of the development of more porous structures, which in turn improves enzyme hydrolysis and significantly changes some physicochemical properties such as gelatinization parameters, depending on the starch type.

Gel hardness (firmness) is the measure of the force needed to deform the gel during compression and is related to its structure and strength. Legume starches (C.P. and T.B.) exhibited the firmest gels among the five tested starches 677 and 481 (g), respectively, whereas after annealing, S.P.S. was the softest 80 (g), but W.S. gel was the softest in the presence of GSE 98 (g) (Table 4), although firm gels are usually associated with high amylose content which is positively correlated with strong amylose network. Therefore, C.P. has high amylose content (24%) which can explain the firm gel, but T.B. (21%) firm gel could be related to the molecular structure (chain length) of its amylose that facilitated for available hydrogen bonding. These two starches exhibited the highest setback as well, which is consistent with their firmer gels, even though the setback is measured at 50 °C and the firmness at room temperature. Regarding S.P.S. (22.4% amylose), the highest peak viscosity and low gel firmness could be accredited to the short amylose chain length which allows for strong packing in the amorphous region of the granule, but when the granule ruptured after gelatinization, these short amylose chains did not form strong gel due to lack of proximity between the molecules. This is true for W.S. where the high amylose content (25%) did not produce the firmest gel, which can also be attributed to the amylose chain length. The cohesiveness is defined as the strength of hydrogen bonding of the gel and the degree to which food gels can be deformed before it breaks. The cohesiveness data did not follow a specific pattern, where firmer gels exhibited low cohesiveness indicating weaker hydrogen bonding. Annealed samples exhibited cohesiveness as follows: C.S. > C.P. > S.P.S. > T.B. > W.S.; however, after GSE treatment, it was W.S. > C.P. > C.S. > T.B. > S.P.S., which indicates significant effect of annealing in GSE (Table 5). W.S gels at 40 and 50 °C had the most deformation ranging between 0.59 and 0.79 before they broke, which indicates strong hydrogen bonding between the gel structures, but at 60 °C, the gel was liquefied. Moreover, the molecular structure and entanglement of other starch gels appear to have been rearranged during the overnight storage. Legume starches, the hardest of the tested starches, appeared to maintain the rank regardless of GSE. Unlike C.P., T.B. deformed easier than C.P. (0.44 versus 0.54 (g)). Generally, harder protein-based gels exhibit higher cohesiveness, but this pattern is not observed here. Likely, storage overnight at room temperature caused molecular rearrangement which was dependent on amylose and amylopectin chain length as well as the granule structure and the degree of molecular entanglement right after gelatinization.

## 4. Conclusions

Wheat starch was the most heat susceptible and liquefied after annealing at 60 °C without GSE. It was also the most swollen at lower annealing temperatures, and in the presence of GSE, the weakest gel was formed at higher temperatures. Sweet potato starch was the least susceptible to heat and GSE, the least swollen, and exhibited the highest peak viscosity. The native C.P. starch presented highest swelling power before annealing and after annealing at 60 °C for 60 min. All tested starches exhibited lower peak viscosity and softer gel in the presence of GSE, and the most change was observed for C.P. and W.S. at lower temperatures and S.P.S. at higher temperatures. The most setback was noted for the legume starches (C.P. and T.B.). Amylose content was not the determinant factor of α-amylase digestion of the annealed starches. The combined factors controlling enzyme susceptibility are granule structure and compactness. Although reports in the literature claimed that high amylose content correlated negatively with the swelling power of some starches, the data presented here indicated that high amylose (W.S.) correlate positively with high swelling power and negatively with peak viscosity. High swelling power of W.S. is consistent with faster enzymatic digestion rates and higher temperatures, which is controlled by rapid diffusion of enzymes into loosely packed granular structures even with the high amylose content. However, S.P.S. (lower amylose content) restricted water uptake due to the packed density of the amorphous region compared to W.S. (higher amylose content). Therefore, GSE is a straightforward low-cost starch hydrolyzer that can be used at industrial scale with further improvement of the germination process.

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
