# Peer review of "The Effect of Germinated Sorghum Extract on the Pasting Properties and Swelling Power of Different Annealed Starches"

_polymers, 2020, doi:10.3390/polym12071602_

Round 1

Reviewer 1 Report

Recommendation: Major revisions for acceptance

Comments to Authors: Hesham, Alqah. M. S Alamri, A. A. Mohamed, S. Hussain, A. A. Qasem, M. A. Ibraheem,  I. A. Ababtain

Manuscript Number: polymers-838707-peer-review-v1

Article Type: Article

Article Title:  The Effect of Germinated Sorghum Extract on the Pasting Properties and Swelling Power of Different Annealed Starches

Overview and general recommendation

The introduction provides a sufficient background and contains relevant references to the problem raised.

The methods were presented correctly and chronologically adequate to the conducted research and are adequately described.

The research design is appropriate.

Minor comments:

  • Abstract: (line 9): I recommend put information what kind of starches exactly was studied.
  • Introduction: (line 39): in my opinion should be 72 h not hr.
  • Fig. 1: native starch should be on the fig.1a like a first one, next annealed starch (fig. 1b); both of figures should have the same scale, till 6000cP; legend on the chart is too big.
  • Tables 1-4: inside tables the abbreviations contain periods e.g. S.P.S. etc., but the discussion and summary are different; the abbreviations used should be the same;
  • bibliography should be improve, the titles of journals should be with capital letters e.g. Carbohydrate Polymers not Carbohydrate polymers, etc.
  • minor shortcomings in the text should be improve.

Major comments:

  • The results are clearly presented in the tables but the discussion and conclusions are not supported by the results.

Overall Recommendation

My recommendation is major revisions for acceptance

Author Response

First Reviewer

Minor comments:

  • Abstract: (line 9): I recommend put information what kind of starches exactly was studied.
  • Starches extracted from chickpea (C.P), corn (C.S), Turkish bean (T.B), sweet potato (S.P.S), and wheat starches (W.S). The extracted starches with different amylose content were annealed in excess water and in germinated sorghum extract (GSE) (1.0 g starch / 9 ml water).
  • Introduction: (line 39): in my opinion should be 72 h not hr.
  • 72 hr was changed to 72 h
  • 1: native starch should be on the fig.1a like a first one, next annealed starch (fig. 1b); both of figures should have the same scale, till 6000cP; legend on the chart is too big.
  • Fig 1 was changed where the native was moved to Fig1a and the annealed to Fig1b, the scale was normalized to 6000, and the legend size was fixed
  • Tables 1-4: inside tables the abbreviations contain periods e.g. S.P.S. etc., but the discussion and summary are different; the abbreviations used should be the same;
  • SPS was changed to S.P.S throughout the paper
  • bibliography should be improved, the titles of journals should be with capital letters e.g. Carbohydrate Polymersnot Carbohydrate polymers, etc.
  • The first letter of the name of the journals in the References section was changed to capital letters
  • minor shortcomings in the text should be improve.
  • Text was reviewed and improved

Major comments:

  • The results are clearly presented in the tables but the discussion and conclusions are not supported by the results.
  • The discussion section was carefully reviewed and the conclusions are in line with the presented data.

Reviewer 2 Report

Detailed recommendation:

Key words: please add “extract” to the key words

Abstract: please add some data to the abstract

Lines 77-79: change font size

Methods: the plant materials must be accurately described (origin, type of crop, size of sample, variety, agrotechnical conditions prevailing during the growing season, etc.)

Do you measure dry mater of starches?

Pasting properties parameters must by better described! (means, unit)

Discussion should by improved.

Author Response

2nd reviewer

Overall Recommendation

My recommendation is major revisions for acceptance

Comments and Suggestions for Authors

Detailed recommendation:

Key words: please add “extract” to the key words

Was done

Abstract: please add some data to the abstract

The following data was added to the abstract. “in addition to the swelling power.  These starches exhibited pasting properties as evidenced by increased peak viscosity with annealing time 3589, 4174, and 4689 cP for annealing at 40, 50, and 60⁰C, respectively. And “The most starch gel setback was recorded for the legume starches, chickpea and Turkish beans (CP 2553 cP and TB 1172 cP)”.

Lines 77-79: change font size

Fund size was adjusted

Methods: the plant materials must be accurately described (origin, type of crop, size of sample, variety, agrotechnical conditions prevailing during the growing season, etc.)

The raw material was purchased from a local supermarket in Riyadh (Saudi Arabia).  The sweet potato used was Salem variety grown in Altayef area of the western Saudi Arabia in the winter, whereas wheat was a hard red winter wheat grown in the north part of Saudi Arabia, the Hail region, in the winter.  The chickpea and Turkish bean were purchased from the local market (Riyadh, Saudi Arabia)     

Do you measure dry mater of starches?

Yes, we did practice dry basis, specifically for weighting the starch for RVA and other test. 

Pasting properties parameters must by better described! (means, unit). 

The pasting parameters of the peak viscosity and setback were properly adjusted and the units were put in order.

Discussion should by improved.

The discussion was reviewed and improved.  The changes are marked with red 

Round 2

Reviewer 1 Report

Recommendation: Minor revisions for acceptance

Comments to Authors: Hesham, Alqah. M. S Alamri, A. A. Mohamed, S. Hussain, A. A. Qasem, M. A. Ibraheem,  I. A. Ababtain

Manuscript Number: polymers-838707-peer-review-v2

Article Type: Article

Article Title:  The Effect of Germinated Sorghum Extract on the Pasting Properties and Swelling Power of Different Annealed Starches

Major comments:

  • The results are clearly presented in the tables but the discussion and conclusions are not supported by the results.

Still the discussion does not have results. For example line 264 “In the most part, annealed C.P exhibited the highest setback (result) and S.P.S and W.S the least (results…), but in the presence of GSE, W.S had by far the least setback (Table 2). These two starches (C.P and W.S) contain the highest amylose content (results…??) amongst the tested starches

I recommend in the discussion in the brackets put the information about values foe example like I have mentioned above. Please make corrections throughout the discussion.

Overall Recommendation

My recommendation is minor revisions for acceptance

Round 2

Response to reviewer 1

  • The results are clearly presented in the tables but the discussion and conclusions are not supported by the results.

Still the discussion does not have results. For example, line 264 “In the most part, annealed C.P exhibited the highest setback (result) and S.P.S and W.S the least (results…), but in the presence of GSE, W.S had by far the least setback (Table 2). These two starches (C.P and W.S) contain the highest amylose content (results…??) amongst the tested starches

I recommend in the discussion in the brackets put the information about values for example like I have mentioned above. Please make corrections throughout the discussion

Regarding the comments of the second reviewer, detailed data was inserted into the discussion section that covers the following segments of the discussions:

Peak viscosity, setback, final viscosity, pasting temperature, swelling power, gel hardness and cohesiveness.  The changes were made in blue color.

Reviewer 2 Report

Manuscript now is correct.

Author Response

No comments were brought up be the 2nd reviewer